# Diagnostic Utility of TSSC3 and RB1 Immunohistochemistry in Hydatidiform Mole

**DOI:** 10.3390/ijms24119656

**Published:** 2023-06-02

**Authors:** Wai Kit Chia, Pik Yuen Chia, Nor Haslinda Abdul Aziz, Salwati Shuib, Muaatamarulain Mustangin, Yoke Kqueen Cheah, Teck Yee Khong, Yin Ping Wong, Geok Chin Tan

**Affiliations:** 1Department of Pathology, Faculty of Medicine, Universiti Kebangsaan Malaysia, Bandar Tun Razak 56000, Kuala Lumpur, Malaysia; wk_chia@ppukm.ukm.edu.my (W.K.C.); salwati@ppukm.ukm.edu.my (S.S.); amar@ppukm.ukm.edu.my (M.M.); 2Department of Diagnostic Laboratory Services, Hospital Canselor Tuanku Muhriz, Universiti Kebangsaan Malaysia, Bandar Tun Razak 56000, Kuala Lumpur, Malaysia; 3Department of Pathology, Hospital Umum Sarawak, Kuching 93586, Sarawak, Malaysia; jessiec0917@gmail.com; 4Department of Obstetrics and Gynecology, Faculty of Medicine, Universiti Kebangsaan Malaysia, Bandar Tun Razak 56000, Kuala Lumpur, Malaysia; norhaslinda.abdaziz@ppukm.ukm.edu.my; 5Department of Biomedical Science, Faculty of Medicine and Health Science, Universiti Putra Malaysia, Serdang 43400, Selangor, Malaysia; ykcheah@upm.edu.my; 6UPM-MAKNA Cancer Research Laboratory, Institute of Bioscience, Universiti Putra Malaysia, Serdang 43400, Selangor, Malaysia; 7Department of Pathology, Women’s and Children’s Hospital, Adelaide, SA 5006, Australia; yee.khong@adelaide.edu.au

**Keywords:** hydatidiform moles, molar pregnancy, TSSC3, RB1, paternal imprinted genes

## Abstract

The general notion of complete hydatidiform moles is that most of them consist entirely of paternal DNA; hence, they do not express p57, a paternally imprinted gene. This forms the basis for the diagnosis of hydatidiform moles. There are about 38 paternally imprinted genes. The aim of this study is to determine whether other paternally imprinted genes could also assist in the diagnostic approach of hydatidiform moles. This study comprised of 29 complete moles, 15 partial moles and 17 non-molar abortuses. Immunohistochemical study using the antibodies of paternal-imprinted (RB1, TSSC3 and DOG1) and maternal-imprinted (DNMT1 and GATA3) genes were performed. The antibodies’ immunoreactivity was evaluated on various placental cell types, namely cytotrophoblasts, syncytiotrophoblasts, villous stromal cells, extravillous intermediate trophoblasts and decidual cells. TSSC3 and RB1 expression were observed in all cases of partial moles and non-molar abortuses. In contrast, their expression in complete moles was identified in 31% (TSSC3) and 10.3% (RB1), respectively (*p* < 0.0001). DOG1 was consistently negative in all cell types in all cases. The expressions of maternally imprinted genes were seen in all cases, except for one case of complete mole where GATA3 was negative. Both TSSC3 and RB1 could serve as a useful adjunct to p57 for the discrimination of complete moles from partial moles and non-molar abortuses, especially in laboratories that lack comprehensive molecular service and in cases where p57 staining is equivocal.

## 1. Introduction

Hydatidiform moles (HM) develop as a result of an abnormal fertilisation of a defective ovum. It is one of the gestational trophoblastic diseases that comprises a group of benign and malignant tumours. There is a distinct geographical distribution of HMs. Estimates from studies across the world suggest the incidence of HMs is higher in Asian countries (0.81–4.4 per 1000 live births) compared to Western countries (0.66–1.21 per 1000 live births) [1,2,3,4]. Interestingly, this observation is also in individuals with Asian heritage who live in Western countries [2]. The reported incidence of HMs was highest in the South-East Asia countries, Indonesia, India and Turkey, with incidence ranging from 2 to 12 per 1000 live births [3]. The incidence of HMs in Malaysia was reported as 2.6 per 1000 live births [4]. In contrast, countries in Europe, North America and Oceania have the lowest incidence of HMs in the world, at 0.66 to 1.21 per 1000 live births [1].

HMs are subdivided into complete mole (CM) and partial mole (PM), based on the combination of histomorphology and genetic evaluations [5]. The differentiation between CMs and PMs is clinically important as it could forecast the likelihood of a recurrence and the risk of developing persistent trophoblastic disease and choriocarcinoma [6]. Most laboratories lack molecular service; hence, they depend solely on histomorphology evaluation for HM diagnosis, which is a challenge to the pathologists. The characteristic histological features of HMs are trophoblast hyperplasia and hydropic degeneration of chorionic villi. However, in particular at the earlier stage of the disease, HMs may not have the classic morphological features [7]. Moreover, CMs and PMs may exhibit substantial overlap in histological characteristics, with significant interobserver variability between practising pathologists [8]. Certain non-molar abortuses (NMA) may also display hydropic alterations which resemble the histomorphology of HMs, further complicating the diagnosis [9].

A CM results from the fertilisation of an ovum devoid of maternal DNA by one or two sperms, leading to a diploid conception consisting entirely of paternal DNA [10]. Rarely, as a result of autosomal recessive mutation, a diploid biparental complete mole may occur. In contrast, a PM is typically the consequence of a dispermic conception with a haploid oocyte and two sperms, resulting in a triploid conceptus. This knowledge forms the basis for the diagnosis of HM, using p57 immunohistochemistry, a paternally imprinted and maternally expressed gene, to distinguish between CM and PM [11,12,13]. The p57 immunohistochemistry is widely acceptable as an ancillary test to aid in the pathological diagnosis of HM. The absence of maternal DNA in CMs typically results in complete loss of p57 expression in the hydropic villi. In contrast, a PM expresses p57 antibody as it contains both paternal and maternal genes [14]. Notably, p57 is unable to differentiate between PM (diandric monogynic triploidy) and non-molar abortus (NMA) (biparental diploidy) specimens because both of them contain maternal DNA [15].

The excess of paternally derived DNA in HMs leads to improper expression of imprinted genes, which results in the overgrowth of trophoblastic cells and defective embryonic development [16,17]. There are approximately 123 imprinted genes in humans, of which 38 are paternally imprinted genes and expressed by the maternal allele (Appendix A) [18]. The cyclin-dependent kinase inhibitor 1C (CDKN1C) gene, which encodes the p57 protein, is one of them [8]. The CDKN1C gene, which is paternally imprinted and can be found on human chromosome 11p15.5, is mainly expressed from the maternal allele in most tissues [19].

Paternally imprinted genes are those that are expressed only when inherited from the mother, while the father’s allele is silenced by DNA methylation [20]. This is known as genomic imprinting, and it has a parent-of-origin-specific effect on gene expression [21]. Paternally imprinted genes play an important role in foetal growth and development, particularly in placental development and nutrient transfer to the developing embryo. Developmental disorders and cancer can result from abnormalities in their expression [22]. Abnormal expression of paternally imprinted genes in HMs can lead to abnormal proliferation of trophoblast cells and the formation of this disease [23]. Therefore, disruptions in the normal regulation of paternally imprinted genes, especially those of paternal origin, can lead to significant implications for pregnancy outcomes and maternal health [24].

The aim of this study is to answer whether other paternally imprinted genes could also assist in the diagnostic approach in HMs. Thus, we have selected some of the paternal-imprinted (RB1, TSSC3 and DOG1) and maternal-imprinted (DNMT1 and GATA3) genes and performed immunohistochemical study to determine their expression in CMs, PMs and NMAs.

## 2. Results

### 2.1. Demographic and Clinical Characteristics

The 61 cases in this study were categorised into 29 CMs (p57-/diploid), 15 PMs (p57+/triploid) and 17 NMAs (p57+/diploid), based on the histomorphology findings, fluorescence in-situ hybridisation DNA ploidy study and p57 immunohistochemistry analysis (Figure 1). Table 1 provides information on the demographic and clinical characteristics of the three groups of patients. The mean age of the CM group was 34.9 years, while that of the PM and NMA groups were 31.3 years and 33.7 years, respectively. The distribution of age among the three groups showed that the majority of the patients in the CM and PM groups were in the age range of 20–39 years, while the majority of the patients in the NMA group were 40 years old and above. The difference in the age distribution among the three groups is not statistically significant (*p* = 0.4567).

As for ethnicity, the majority of the patients in all three groups were Malay, with 79.3%, 100% and 88.2% in the CM, PM and NMA groups, respectively. A small proportion of patients in the CM (10.3%) and NMA (11.8%) groups were Chinese, and 10.3% in the CM were of other ethnicities (Punjabi and Caucasian), while none of the patients in the PM group belonged to other ethnicities. The mean gestational age in the CM group was 10.6 weeks, while that in the PM and NMA groups were 13.2 weeks and 10.6 weeks, respectively. The difference in gestational age between the CM and PM groups is statistically significant (*p* = 0.0247). The mean β-hCG levels in the CM, PM and NMA groups were 234,024, 102,174 and 10,124 mIU/mL, respectively. The mean β-hCG levels in the CM was higher than those in the PM, and the difference is statistically significant (*p* = 0.0227).

### 2.2. Immunohistochemical Study of Paternal- and Maternal-Imprinted Genes

#### 2.2.1. Paternally Imprinted Genes

##### TSSC3 Immunohistochemistry

Tumour-suppressing STF cDNA 3 (TSSC3) (also known as Pleckstrin homology-like domain family A member 2/PHLDA2) was expressed in the cytoplasm of cytotrophoblasts (CT) of all cases of PMs (15/15, 100%) and NMAs (17/17, 100%). In contrast, only 31.0% (9/29) of the CMs were positive for TSSC3 (Figure 2). Six of the nine positive cases showed staining of weak intensity. The difference in TSSC3 expression in CMs compared to PMs and NMAs is statistically significant (*p* < 0.0001). The villous stromal cells (VSC) were mostly negative for TCCS3, except for two cases of PMs. Syncytiotrophoblasts (ST) were consistently negative for TSSC3 in all categories. The decidual cells (DC) were mostly positive, i.e., 23/24 (95.8%), 12/15 (80%) and 12/15 (80%), in CMs, PMs and NMAs, respectively. On the other hand, the intermediate trophoblasts (IT) were predominantly negative; TSSC3 expression was observed in 2/29 (6.9%), 5/15 (33.3%) and 4/16 (25%) in CMs, PMs and NMAs, respectively. TSSC3 was not expressed in the nuclei of all types of cells in the placenta. Table 2 shows the distribution of protein expression of paternal- and maternal-imprinted genes in different types of placental cells in CMs, PMs and NMAs.

##### RB1 Immunohistochemistry

RB transcriptional corepressor 1 (RB1) expression was observed in the nuclear of CTs, VSCs, ITs and DCs (Figure 2). None of the STs showed RB1 staining. In CTs, RB1 was positive in 3 of the 29 cases of CMs (10.3%). In contrast, RB1 was positive in all cases of PMs (15/15, 100%) and NMAs (16/17, 94.1%). The difference in RB1 expression in CMs compared to PMs and NMAs is statistically significant (*p* < 0.0001). In VSCs, only one case from the CM group was positive with moderate intensity staining, while all cases in the PM and NMA groups were negative. ITs were consistently negative for RB1 staining. The DCs in the majority of cases were positive for RB1, 88% (22/25), 86.7% (13/15) and 92.9% (13/14), in CMs, PMs and NMAs, respectively.

##### DOG1 Immunohistochemistry

We performed a preliminary staining on 12 cases of each group (CM, PM and NMA) and delay of germination 1 (DOG1) was consistently negative in all cell types in the placenta (Figure 2).

#### 2.2.2. Maternally Imprinted Genes

##### DNMT1 Immunohistochemistry

DNA methyltransferase 1 (DNMT1) was expressed in the nuclei of CTs in all cases of CMs, PMs and NMAs (Figure 2). It showed variable staining in other cell types in the placenta. The majority of ITs were positive for DNMT1, i.e., 93.1% (27/29), 93.3% (14/15) and 76.8% (13/17) of cases of CMs, PMs and NMAs, respectively. It was positive in 41.4% (12/29), 33.3% (5/15) and 11.8% (2/17) of the VSCs in CMs, PMs and NMAs, respectively. In STs, DNMT1 was positive in 51.7% (15/29), 40% (6/15) and 41.2% (7/17) of cases of CMs, PMs and NMAs, respectively. In DCs, it was positive in 56.5% (13/23), 53.3% (8/15) and 61.5% (8/13) of cases of CMs, PMs and NMAs, respectively. There is no statistically significant difference in DNMT1 expression in all types of placental cells between CMs and PMs (*p* > 0.5) (Table 3).

##### GATA3 Immunohistochemistry

GATA binding protein 3 (GATA3) was expressed in the nuclei of all CTs and STs in all three groups, except for one case of CM where the staining was negative (Figure 2). There is no statistically significant difference between the three groups (*p* = 1.0). GATA3 was consistently positive in all ITs. In contrast, it was negative in VSCs of all three groups. In STs, GATA3 was positive in almost all cases in the three groups, except for two cases of CMs where the staining was negative. The DCs showed variable staining, i.e., 4.2% (1/24), 53.3% (8/15) and 13.3% (2/15) in CMs, PMs and NMAs. The difference in GATA3 staining in DCs between CMs and PMs is statistically significant (*p* = 0.0008).

### 2.3. TSSC3 and RB1 Immunohistochemistry in Equivocal Cases

We were unable to ascertain the diagnosis of four cases in this study by histomorphology alone. However, after the utility of p57 immunohistochemistry and DNA ploidy study, we categorised two of them (PM002 and PM019) as non-molar abortus and one as partial mole (PM009) (Table 4). PM003 was uncertain as p57 staining was only 50% (equivocal), and it was diploid by FISH analysis. The addition of a positive TSSC3 immunohistochemistry implied that it is likely a case of non-molar abortus, with rare exception of a diploid biparental CM and retention of maternal DNA. While in the other three cases, TSSC3 and RB1 reinforced the diagnosis.

## 3. Discussion

Our study found about a quarter of the subjects with CM were >40 years old. In contrast, none of the subjects with PM was >40 years old. Previous studies reported HM occurrence has a bimodal distribution with subjects <20 years and >40 years old [19,25,26]. Women after the age of 35 years old have a 5–10 times higher risk for developing HM [27]. Women with a history of previous molar pregnancy have a 10 times higher risk than sporadic cases in developing HMs [28].

The β-hCG level in the CM group was at the average range of about 200,000 mIU/mL, while the PM group was at about 100,000 mIU/mL. This finding is similar to our previous study [11]. Although the level of β-hCG in the PM group was higher than NMA (about 10,000 mIU/mL), the level was still within the range of normal pregnancy [29]. Nonetheless, raised serum β-hCG should be interpreted with caution as it can be elevated in β-hCG-secreting tumours, and a low level of β-hCG may be due to infarcted HM [30].

There are substantial overlapping histomorphological features in distinguishing CM from PM, which could potentially lead to misdiagnosis and subsequent inappropriate clinical management [8]. Our previous study reported that the diagnostic accuracy of histomorphology alone was 78.4% for CMs and 70.6% for PMs [5]. Notably, the diagnosis based on histomorphology is subjected to inter- and intra-observer variabilities, resulting in suboptimal diagnostic accuracy and reproducibility [19].

The use of ancillary techniques like immunohistochemistry, FISH, karyotyping, flow cytometry or genotyping had been shown to improve the diagnostic accuracy of HMs [5,19,31,32]. P57, a paternally imprinted maternally expressed gene, is currently a widely used immunohistochemistry method in the diagnosis of HMs. As, generally, CMs do not have maternal genetic material, p57 expression is expected to be negative in CMs. On the contrary, p57 should be positive in villous cytotrophoblasts of biparental conceptus like PMs and NMAs [11,31]. Meanwhile, most CMs are diploid, and most PMs are diandric triploid with two paternal and one maternal genome [32]. Therefore, DNA ploidy study can also assist in the differentiation between CMs and PMs. All the cases in our study have been separated into three groups, i.e., CM (p57-/diploid), PM (p57+/triploid) and NMA (p57+/diploid) using a combination of p57 immunohistochemistry and DNA ploidy study by FISH.

Genomic or parental imprinting is an epigenetic modification of the genome in the gametes that causes an imbalanced expression of maternal versus paternal alleles in the offspring. Studies have shown about half of them control tissue growth [33]. There are about 38 known paternally imprinted maternally expressed genes. We were curious whether there are other paternally imprinted genes that could act similarly to p57 in the diagnosis of HMs. Based on the parental conflict theory, the paternally imprinted maternally expressed genes are essential for embryonic development, while the maternally imprinted paternally expressed genes are essential for placental development. [34]. A study using animal models with deficient maternally or paternally imprinted genes resulted in a larger or smaller conceptus [35].

TSSC3 is located in chromosome 11p15.5 that contains several other well-studied imprinted genes, including H19, IGF2, ASCL2 and p57 [36]. This gene was the first apoptosis-related gene that was shown to be imprinted [37], and it has been established to regulate placental growth [38,39]. The placenta is an organ with high amounts of the TSSC3 gene, and it persists throughout gestation in the human placenta [40]. Salas et al. (2004) reported that loss of TSSC3 resulted in placentomegaly, while its overexpression led to stunted placental growth [33].

We found about 69% of the CMs were negative for TSSC3 in the villous cytotrophoblasts. In contrast, all PMs and NMAs expressed TSSC3. Studies by Thaker et al. (2004), Kato et al. (2005) and Saxena et al. (2003) showed complete loss of TSSC3 expression in all cases of CMs [36,40,41]. Intriguingly, nine cases of CMs in our study were positive for TSSC3, of which the majority (6/9, 66.7%) had weak staining intensity (Appendix A). Fisher et al. (2004) demonstrated that a retention of a maternal copy of chromosome 11, which encodes for both p57 and TSSC3, may occur [26]. P57 (CDKN1C, size 2559) and TSSC3 (PHLDA2, size 1448) are located close to each other in chromosome 11. Therefore, co-retention of these two genes might happen. McConnell et al. (2009) also described an example of androgenetic diploidy CM with retained maternal copies of chromosomes 6 and 11, with abnormal p57 expression [16]. We postulated that some DNA of maternal origin may be retained in CMs; hence, the positive expression of TSSC3 in some of our cases. Further studies are needed to ascertain this finding.

The human retinoblastoma gene (RB1) is located at chromosome 13q14.2. It was the first cloned tumour suppressor gene to be expressed preferentially from the maternal allele [42,43]. RB1 functions as a negative regulator of the cell cycle, to maintain a balance between cell growth and development by binding to transcription factors and regulating the expression of genes involved in cell proliferation and differentiation. The role of RB1 includes the regulation of cell cycle, cell senescence, growth arrest, apoptosis and differentiation [44]. We found only 3 of the 29 cases (10.3%) of CMs was positive for RB1, as compared to PMs and NMAs, where all cases expressed strong immunopositivity. The p57 gene is responsible for encoding a cyclin-dependent kinase inhibitor that functions upstream of the RB1 protein [45]. As both p57 and RB1 are involved in the same cell cycle pathway, this suggests maternal suppression of cell proliferation may have been subject to evolutionary selection [42,43].

We performed a preliminary staining of DOG1 in 12 cases of each category and found no immunoreactivity in all placenta cells. This finding is consistent with previous observation that DOG1 was not found in the placenta, thymus, tonsil, testis and postmenopausal ovarian tissues [46]. DOG1 immunopositivity was observed in the adult gastrointestinal tract, salivary glands, pancreatic acini, intrahepatic bile ducts, gallbladder glandular elements, breast myoepithelial cells and prostatic basal cells [47].

GATA3 and DNMT1 are maternally imprinted, paternally expressed genes, and both of them showed variable positivity in all samples, except for GATA3 in VSCs, where it was negative. Studies reported GATA3 was frequently expressed in trophoblastic tumours such as choriocarcinoma, placental site trophoblastic tumour, epithelioid trophoblastic tumour and hydatidiform mole [48,49]. DNMT1 is involved in supporting fetoplacental growth [50,51].

## 4. Materials and Methods

### 4.1. Formalin-Fixed Paraffin-Embedded Hydatidiform Mole Tissue Samples

This is a cross-sectional study comprising a total of 61 formalin-fixed paraffin-embedded (FFPE) tissue sections diagnosed as hydatidiform mole and hydropic abortus (29 cases of CMs, 15 PMs and 17 NMAs), collected between the year 2015 and 2021. The diagnosis of all cases was confirmed based on histological examination, p57 immunohistochemistry and fluorescence in-situ hybridisation DNA ploidy study [5]. The haematoxylin and eosin-stained slides were reviewed by pathologists (G.C.T and Y.P.W.), and one FFPE tissue block that was most representative of the lesion was selected for immunohistochemical study. The tissue blocks were retrieved from the archive of the Department of Pathology. Ethics approval was granted by our institutional human ethics committee (approval code: JEP-2019-820).

### 4.2. DNA Ploidy Study by Fluorescence In Situ Hybridization

FISH was performed on formalin-fixed paraffin embedded tissue sections using the Dako Histology FISH Accessory Kit (Glostrup, Denmark) according to the manufacturer’s protocol. Tissue sections (4 m) were baked at 62 °C for 15 min before being deparaffinized in xylene and rehydrated in a series of ethanol dilutions. The sections were then pretreated in Pre-Treatment Solution, protein digested with Pepsin Solution for 30 min at 37 °C, and digestion adequacy was determined using 4′,6-diamidino-2-phenylindole (DAPI) [52]. For each case, two sets of probes were used: CytoCell^®^ Satellite Enumeration Probes (Cambridge, UK) for (a) chromosome 11 (green) and 16 (red), and (b) chromosome X (green) and Y (red). Sections were then denatured for 5 min at 76 °C before being incubated overnight at 37 °C in a ThermoBrite FISH Slide Processing System (Leica Biosystems, Richmond, IL, USA). Following the washing and rinsing steps with Stringent Wash Buffer, 6 µL of DAPI was applied to each section and cover-slipped. A fluorescence microscope was used to examine all of the slides. A total of 250 cells were counted from five chorionic villi (50 cells from each), and the ploidy status was determined as diploid or triploid if more than 10% of nuclei showed two or three signals, respectively.

### 4.3. Immunohistochemistry of Paternally and Maternally Imprinted Genes

Antibodies of four paternal-imprinted genes (p57, RB1, TSSC3, DOG1) and two maternal-imprinted genes (DNMT1 and GATA3) were selected to perform the immunohistochemical study on CMs, PMs and NMAs. All immunohistochemistry studies were conducted using the EnVisionTM FLEX Mini Kit, High pH (Dako, Denmark), following the manufacturer’s guides and recommendations. In general, the FFPE tissue sections were sectioned at 4 μm in thickness and mounted on poly-L-lysine coated glass slides and subsequently baked at 62 °C overnight. The tissue sections were deparaffinised and rehydrated before performing antigen retrieval with target retrieval solution, citrate pH 6.0 and pH 9.0 (Dako, Denmark) for TSSC3 and the other antibodies, respectively, in the Decloaking Chamber™ NxGen (Biocare Medical, Pacheco, CA, USA) for 30 min at 110 °C. Next, they were treated with peroxidase-blocking reagent (Dako, Denmark) for 10 min to block endogenous peroxidase activity. All sections were incubated with the primary antibodies along with their respective positive controls and dilution factors, as listed in Table 5. Lastly, the slides were incubated with EnVision^TM^ FLEX HRP (Dako, Denmark), followed by 3,3′-diaminobenzidine-containing chromogenic (DAB) stain and haematoxylin counterstain, before mounted with coverslips and DPX mounting medium.

Immunoexpression analyses of all the immunostained sections were independently examined by two pathologists (G.C.T and Y.P.W) blinded from the original diagnosis. The immunoreactivity of antibodies was assessed on the following cell types in the placenta, i.e., cytotrophoblasts, syncytiotrophoblasts, villous stromal cells, intermediate trophoblasts and decidual cells. The nuclear immunoreactivity for DNMT1 and GATA3 antibodies and cytoplasmic immunoreactivity for TSSC3 and DOG1 antibodies of any intensities detected in ≥10% of cells were considered as positive. For RB1, staining of <50% in the nuclei of cytotrophoblasts was considered as negative, while staining of ≥50% was considered as positive.

### 4.4. Statistical Analysis

Statistical analysis was carried out using GraphPad Prism (version 8.4.0 for MacOS, GraphPad Software, San Diego, CA, USA). Demographic and clinical characteristic data were analysed with Student’s *t*-test, while immunohistochemistry scoring data were evaluated with a chi-square test. A *p* value of <0.05 was considered as statistically significant.

## 5. Conclusions

The loss of TSSC3 promotes placental growth, and the loss of p57/RB1 increases cell proliferation. These may represent the pathogenesis of hydatidiform moles. The diagnosis of hydatidiform moles can be challenging without the support of an ancillary test. TSSC3 and RB1 could be used as an adjunct in the diagnosis of hydatidiform moles, particularly when p57 immunohistochemistry provides an equivocal result.

## Figures and Tables

**Figure 1 ijms-24-09656-f001:**
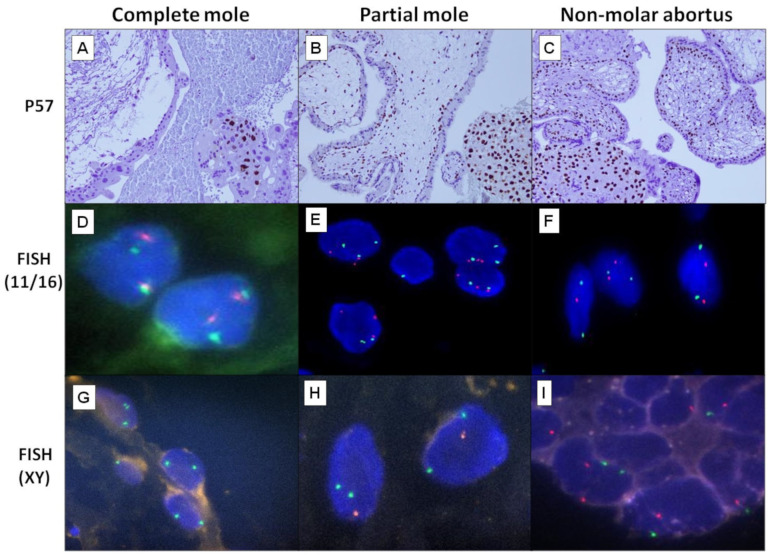
p57 immunohistochemistry and FISH DNA ploidy analysis. Our cases were categorised into complete mole, partial mole and non-molar abortus by histomorphology, p57 and ploidy studies. (**A**) In complete mole, there is loss of p57 expression in the cytotrophoblast cells (×20). In contrast, strong p57 expression was observed in the nuclei of cytotrophoblast cells of (**B**) partial mole (×20) and (**C**) non-molar abortus (×20). FISH using centromere 11 (green) probe and centromere 16 (red) probe showed the presence of two green and two red signals (diploid) (×100) (**D**) complete mole and (**F**) non-molar abortus, while in partial mole (**E**), three green and three red signals (triploid) were observed. FISH using centromeric X (green) and centromeric Y (red) probes showed two green signals, indicating XX in complete mole (**G**). (**H**) Partial mole showed two green signals and one red signal pattern (XXY). (**I**) Non-molar abortus showed one green and one red signal (XY).

**Figure 2 ijms-24-09656-f002:**
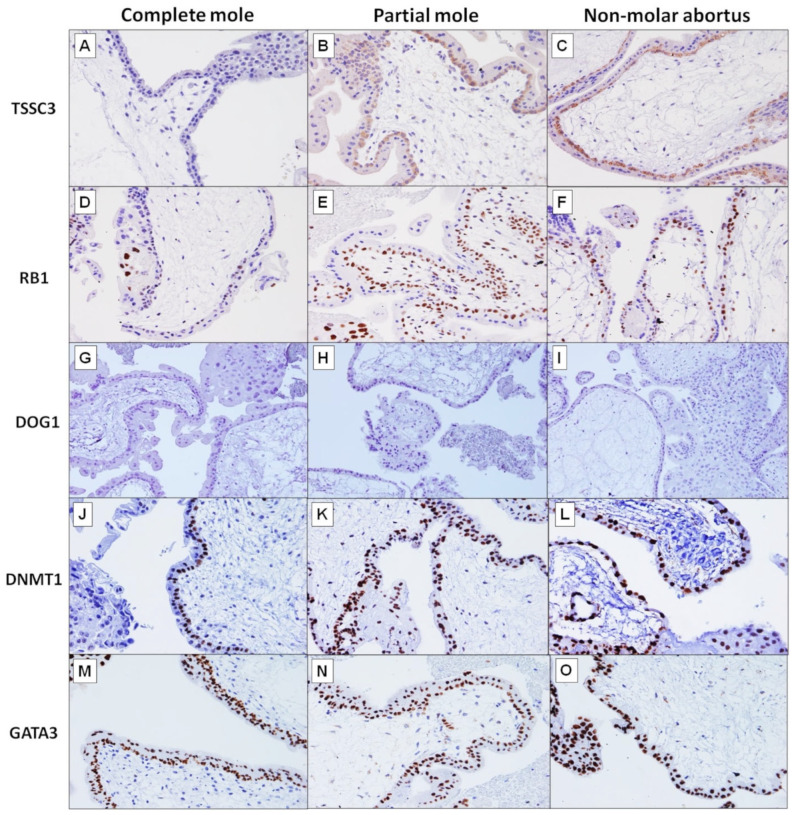
The immunohistochemistry profile of paternal- and maternal-imprinted genes (×20). There was no TSSC3 expression in the cytoplasm of cytotrophoblasts of complete mole (**A**). In contrast, both partial mole (**B**) and non-molar abortus (**C**) showed strong cytoplasmic TSSC3 expression in cytotrophoblasts. RB1 was weakly positive in nuclei of cytotrophoblasts in complete mole (**D**). However, it showed strong immunoreactivity in the nuclei of cytotrophoblasts of both partial mole (**E**) and non-molar abortus (**F**). DOG1 was not expressed in complete mole (**G**), partial mole (**H**) and non-molar abortus (**I**). Strong DNMT1 immunoreactivity was seen in the nuclei of cytotrophoblasts of complete mole (**J**), partial mole (**K**) and non-molar abortus (**L**). Strong GATA3 immunoreactivity was seen in the nuclei of cytotrophoblasts of complete mole (**M**), partial mole (**N**) and non-molar abortus (**O**).

**Table 1 ijms-24-09656-t001:** Demographic and clinical characteristics of complete mole, partial mole and non-molar abortus in this study.

Demographic	CM *n* (%)	PM *n* (%)	NMA *n* (%)	*p* Value CM vs. PM
Age range (years old)	21–51	24–38	21–42	
Mean (SD)	34.9 (8.841)	31.3 (4.03)	33.7 (6.828)	*p* = 0.4567
20–29 years old	9 (31.0%)	5 (33.3%)	5 (29.4%)	
30–39 years old	12 (41.4%)	10 (66.7%)	7 (41.2%)	
40 years old and above	8 (27.6%)	0 (0%)	5 (29.4%)	
Gestational age range (weeks)	7–15	9–18	8–12	
Mean (SD) ^a^	10.60 (2.415)	13.17 (3.125)	10.57 (1.284)	*p* = 0.0247 *
β-hCG levels range (mIU/mL)	22,683–1,000,000	6985–369,109	<1.2–25,888	
Mean (SD) ^b^	234,024 (250,856)	102,174 (135,505)	10,124 (12,563)	*p* = 0.0227 *
Ethnicity				
Malay	23 (79.3%)	15 (100%)	15 (88.2%)	
Chinese	3 (10.3%)	0 (0%)	2 (11.8%)	
Others (Punjabi, Caucasian)	3 (10.3%)	0 (0%)	0 (0%)	

CM—Complete mole, PM—Partial mole, NMA—Non-molar abortus, ^a^ Number of cases in gestational age: 25 CM, 12 PM and 14 NMA, ^b^ Number of cases in β-hCG levels: 24 CM, 10 PM and 4 NMA, * *p* value of <0.05 is considered as statistically significant.

**Table 2 ijms-24-09656-t002:** Expressions of paternally imprinted genes (TSSC3 and RB1) of the various placental cell types in complete mole, partial mole and non-molar abortus.

TSSC3		CM	PM	NMA	*p* Value CM vs. PM	*p* Value CM vs. NMA
CT	Positive	9	15	17	<0.0001 *	<0.0001 *
	Negative	20	0	0		
VSC	Positive	0	2	0	0.111	1.0
	Negative	29	13	17		
ST	Positive	0	0	0	1.0	1.0
	Negative	29	15	17		
IT	Positive	2	5	4	0.1656	0.0358
	Negative	27	10	12		
DC	Positive	23	12	12	0.2785	0.2785
	Negative	1	3	3		
**RB1**						
CT	Positive	3	15	16	<0.0001 *	<0.0001 *
	Negative	26	0	1		
VSC	Positive	1	0	0	1.0	1.0
	Negative	28	15	17		
ST	Positive	0	0	0	1.0	1.0
	Negative	29	15	17		
IT	Positive	29	15	17	1.0	1.0
	Negative	0	0	0		
DC	Positive	22	13	13	1.0	1.0
	Negative	3	2	1		

CM—complete mole, PM—partial mole, NMA—non-molar abortus, CT—cytotrophoblast, VSC—villous stromal cell, ST—syncytiotrophoblast, IT—intermediate trophoblast, DC—decidual cell, * *p* value of <0.05 is considered as statistically significant.

**Table 3 ijms-24-09656-t003:** Expression of maternally imprinted genes (DNMT1 and GATA3) of the various cell types in complete mole, partial mole and non-molar abortus.

DNMT1		CM	PM	NMA	*p* Value CM vs. PM	*p* Value CM vs. NMA
CT	Positive	29	15	17	1.0	1.0
	Negative	0	0	0		
VSC	Positive	12	5	2	0.7477	0.0487 *
	Negative	17	10	15		
ST	Positive	15	6	7	0.535	0.552
	Negative	14	9	10		
IT	Positive	27	14	13	1.0	0.1744
	Negative	2	1	4		
DC	Positive	13	8	8	1.0	1.0
	Negative	10	7	5		
**GATA3**						
CT	Positive	28	15	16	1.0	1.0
	Negative	1	0	0		
VSC	Positive	0	0	0	1.0	1.0
	Negative	29	15	17		
ST	Positive	27	15	16	0.5402	0.5313
	Negative	2	0	0		
IT	Positive	29	15	17	1.0	1.0
	Negative	0	0	0		
DC	Positive	1	8	2	0.0008 *	0.547
	Negative	23	7	13		

CM—complete mole, PM—partial mole, NMA—non-molar abortus, CT—cytotrophoblast, VSC—villous stromal cell, ST—syncytiotrophoblast, IT—intermediate trophoblast, DC—decidual cell, * *p* value of <0.05 is considered as statistically significant.

**Table 4 ijms-24-09656-t004:** The utility of TSSC3 and RB1 in establishing the diagnosis of equivocal cases.

Sample No.	PM002	PM003	PM009	PM019
Maternal age (years)	30	30	38	27
Gestational age (weeks)	NA	8	18	Not available
β-hCG level (mIU/mL)	184,108.50	181,113.20	33,906	Not available
Preliminary diagnosis based on histomorphological features alone	? Partial mole	Uncertain	? Partial mole	? Partial mole
p57 immunohistochemistry (Percentage of positivity)	100%	50%	60%	100%
DNA Ploidy Status	2n, XX	2n, XX	3n, XXX	2n, XX
Diagnosis (histomorphology + p57 + DNA ploidy study)	Non-molar abortus	Uncertain	Partial mole	Non-molar abortus
TSSC3 immunohistochemistry (Intensity/percentage)	Positive 3+/20%	Positive 2+/50%	Positive 3+/50%	Positive 3+/40%
RB1 immunohistochemistry (Percentage)	Positive 50%	Negative 40%	Positive 100%	Positive 100%
Final diagnosis	Non-molar abortus	Non-molar abortus	Partial mole	Non-molar abortus
TSSC3 and RB1 remarks	Reinforced the diagnosis	The p57 staining was equivocal in this case. TSSC3 showed 50% positivity; it is likely a non-molar abortus. Except for the rare possibility of a diploid biparental complete mole or a retention of maternal DNA.	Reinforced the diagnosis	Reinforced the diagnosis

**Table 5 ijms-24-09656-t005:** List of antibodies of paternally and maternally imprinted genes.

Primary Antibodies [Clone Code]	Manufacturer	Catalog Number	Imprinted Gene	Expression Site	Control	Dilution Factor
P57 Rabbit monoclonal [EP2515Y]	Abcam, Cambridge, UK	AB75974	Paternal	Nucleus	Placenta	1:500
RB Rabbit monoclonal [EPR17512]	Abcam, Cambridge, UK	AB181616	Paternal	Nucleus	Tonsil	1:1500
TSSC3 Rabbit polyclonal	Abcam, Cambridge, UK	AB234669	Paternal	Cytoplasm	Placenta	1:300
DOG1 Rabbit monoclonal (SP31)	Cell Marque, California, USA	244R-14	Paternal	Cytoplasm	GIST	1:500
DNMT1 Rabbit monoclonal [EPR18453]	Abcam, Cambridge, UK	AB188453	Maternal	Nucleus	Tonsil	1:100
GATA3 Mouse monoclonal (L50-823)	Cell Marque, California, USA	390M-14	Maternal	Nucleus	Placenta	1:500

GIST—gastrointestinal stromal tumour, UK—United Kingdom, USA—United States of America.

## Data Availability

All our data already provided in the supplementary materials.

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
