# Peer review of "Diagnostic Utility of TSSC3 and RB1 Immunohistochemistry in Hydatidiform Mole"

_ijms, 2023, doi:10.3390/ijms24119656_

Round 1

Reviewer 1 Report

In this paper authors conducted a study on three gestational diseases : Complete mole (CM),– Partial mole(PM) and Non-molar abortus(NMA)  using antibodies of paternal-imprinted (RB1, TSSC3, and DOG1) and maternal-imprinted (DNMT1 and GATA3) genes and established TSSC3 and RB1 could serve as a useful adjunct to  p57 for the discrimination of complete mole from partial mole and non-molar abortus.  

Based on their data few questions need to answer.

Was the three patients showing positivity to RB1 in CM also positive to TSSC3?

Did you conduct any correlation study between the level of β-hCG and TSSC3 and RB1 in all patients and if yes was there any?

Author Response

Reviewer 1

In this paper authors conducted a study on three gestational diseases: Complete mole (CM),– Partial mole (PM) and Non-molar abortus (NMA) using antibodies of paternal-imprinted (RB1, TSSC3, and DOG1) and maternal-imprinted (DNMT1 and GATA3) genes and established TSSC3 and RB1 could serve as a useful adjunct to p57 for the discrimination of complete mole from partial mole and non-molar abortus.  

Based on their data few questions need to answer.

1) Was the three patients showing positivity to RB1 in CM also positive to TSSC3?

Response: Thank you for the comments. 

Of the 3 cases of CM with RB1 positivity, 2 of them were negative toward TSSC3 and one was positive. See supplementary table 2.

No.

Sample No.

βhCG (mIU/mL)

Diagnosis

TSSC3

RB1

Intensity

%

Intensity

%

1

CM001

170,624

CM

-

0

1+

30

2

CM004

583,933

CM

1+

100

1+

30

3

CM005

170,000+

CM

-

0

1+

30

4

CM006

NA

CM

-

0

2+

20

5

CM007

67,899.50

CM

-

0

-

0

6

CM008

186,596

CM

-

0

2+

30

7

CM009

542,643

CM

-

0

2+

30

8

CM010

670,773.80

CM

-

0

2+

10

9

CM011

1,000,000

CM

1+

20

2+

40

10

CM012

200,382

CM

-

0

2+

30

11

CM013

25,000

CM

-

0

2+

30

12

CM014

34,985.90

CM

-

0

2+

33

13

CM016

187,479.50

CM

-

0

2+

20

14

CM017

NA

CM

1+

20

2+

20

15

CM018

NA

CM

-

0

1+

30

16

CM019

NA

CM

-

0

2+

100

17

CM020

NA

CM

-

0

2+

30

18

PM002

184,108.50

CM

3+

20

2+

50

19

PM003

181,113.20

CM

2+

50

2+

40

20

PM005

NA

CM

1+

10

2+

30

21

PM006

NA

CM

1+

10

1+

10

22

PM007

400,000

CM

1+

30

2+

20

23

PM011

NA

CM

-

0

2+

10

24

PM012

79,677

CM

-

0

2+

30

25

PM014

89,116

CM

2+

80

2+

10

26

PM015

131,942

CM

-

0

2+

40

27

PM016

79,284

CM

-

0

2+

30

28

PM021

NA

CM

-

0

2+

10

29

PM022

NA

CM

-

0

2+

60

2) Did you conduct any correlation study between the level of β-hCG and TSSC3 and RB1 in all patients and if yes was there any?

Response: Thank you for pointing out this.

There were a total of 19 cases of CM where the β-hCG levels were available, while PM and NMA both have 9 and 4 cases with data on β-hCG.

Of the 19 cases of CM, 6 were positive for TSSC3 and 13 cases were negative. The average level of β-hCG in positive and negative TSSC3 expressions were 406378 and 198107, respectively. There is no statistical difference in β-hCG between positive and negative TSSC3 (p = 0.1116, t-test). See table below. 

As for RB1, there was only one positive case in CM, where the β-hCG was available, and 18 were negative. The average level of β-hCG in positive and negative RB1 expressions were 184108 and 272438, respectively. Unable to perform a statistical analysis as there was only one case of CM.

Complete mole: β-hCG in TSSC3

b-hCG

TSSC3 Positive

TSSC3 Negative

Mean

406378

198107

SD

341981

202074

SEM

139613.2

56045.2

N

6

13

P = 0.1116

Complete mole: β-hCG in RB1

b-hCG

RB1 Positive

RB1 Negative

Mean

184,108.50

272,438

N

1

18

Reviewer 2 Report

The manuscript "Diagnostic utility of TSSC3 and RB1 immunohistochemistry in hydatidiform mole" is an interesting manuscript on the role of TSSC3 and RB1 immunohistochemistry in hydatidiform mole. The work is original and well structured, giving important novelty to scientific literature. The design of the project is appropriate and the results are significant. The number of patients is not so high, but, considering the rare frequency of this clinical condition, it is appropriate. The statistical analysis is well conducted and the language is acceptable.
It represents a valid work and it gives the opportunity to focus attention on a better diagnostic strategy for an early diagnosis of 
hydatidiform mole.

Author Response

Response: Thank you for the comments.

Reviewer 3 Report

the manuscript is interesting, well illustrated and generally well written. The manuscript can be accetped in the present form.

Author Response

Response: Thank you for the comments.
